# Fabrication and Characterization of Curcumin-Complexed Nanoparticles Using Coconut Protein Nanoparticles

**DOI:** 10.3390/pharmaceutics17101247

**Published:** 2025-09-24

**Authors:** Leila Ziaeifar, Maryam Salami, Gholamreza Askari, Zahra Emam-Djomeh, Raimar Loebenberg, Michael J Serpe, Neal M. Davies

**Affiliations:** 1Department of Food Science, Engineering and Technology, College of Agriculture & Natural Resources, University of Tehran, Karaj Campus, Karaj 1417466191, Iran; leila.ziaeifar@ut.ac.ir (L.Z.); iraskari@ut.ac.ir (G.A.); emamj@ut.ac.ir (Z.E.-D.); 2Faculty of Pharmacy and Pharmaceutical Sciences, University of Alberta, Edmonton, AB T6G 2E1, Canada; raimar@ualberta.ca; 3Department of Chemistry, Faculty of Science, University of Alberta, Edmonton, AB T6G 2G2, Canada; serpe@ualberta.ca

**Keywords:** coconut protein, nanoparticle, complexation, antioxidant activity, release kinetic

## Abstract

**Background/Objectives**: Curcumin (Cur) has various biological properties, including anti-microbial, antioxidant, anticancer, anti-diabetic, anticarcinogenic, antitumor, and anti-inflammatory activities. However, using Cur in functional food products is challenging because of its low solubility in an aqueous environment, rapid degradation, and low bioavailability. Nanostructure delivery systems provide a high surface area to volume ratio and sustainable release properties. **Methods**: Coconut protein nanoparticles (CPNPs) have been fabricated through heat treatment at 85 °C and pH 2 for 5 h. The formation of CPNP-Cur was used to improve Cur solubility, followed by antioxidant activity at neutral pH in an aqueous solution. **Results**: The maximum efficiency and loading capacity of Cur in CPNP were 96.6% and 19.32 µg/mg protein, respectively. Scanning electron microscopy indicated the spherical and organized shape of CPNP with a small size of 80 nm. The fluorescence quenching of CPNP-Cur confirmed the potential of Cur to bind to the tryptophane and tyrosine residues in CPNP. The structural properties of CPNP and CPNP-Cur were investigated using FTIR and X-ray diffraction. The antioxidant activity of samples, measured with the ABTS radical scavenging method, demonstrated that the antioxidant capacity of the aqueous solution of Cur was significantly enhanced through the encapsulation into CPNP. The steady release of Cur was observed in the simulated gastrointestinal tract, and the percentage of the cumulative release increased up to 29.2% after 4 h. **Conclusions**: Our findings suggest that CPNP was a suitable nanocarrier for Cur due to improved antioxidant activity and controlled release behavior. These results are valuable for the development of coconut protein nanoparticles to use as a novel nano-delivery system of bioactive components.

## 1. Introduction

Curcumin (Cur) is the main bioactive component extracted from the rhizome of turmeric. It exhibits various biological properties, including antimicrobial, antioxidant, anticancer, anti-diabetic, anticarcinogenic, antitumor, and anti-inflammatory activities [1,2]. However, incorporating Cur into functional food products is challenging due to its low solubility in aqueous environments, rapid degradation, and limited bioavailability [3]. To overcome these issues, various nanostructure delivery systems have been developed to encapsulate Cur, such as nanoemulsions, nanoparticles, liposomes, nanocapsules, and micelles. These systems offer a high surface area-to-volume ratio and sustained release properties [4,5,6,7]. They are primarily constructed using natural polymers like lipids, proteins, and polysaccharides [8]. Food proteins are particularly effective carriers of bioactive compounds because of their amphiphilic nature, high functionality, and excellent nutritional value [9]. Animal proteins (e.g., casein, whey, gelatin, and collagen) are promising nanocarrier materials, according to Wang et al. [10]. Despite their effectiveness, there is a growing trend toward developing protein-rich functional foods from vegan plant-based proteins due to the high cost of animal proteins, environmental concerns, and dietary restrictions stemming from moral or religious beliefs [10,11]. Recently, food-grade nanoparticles have been employed as carriers for bioactive compounds, offering benefits such as low toxicity, biodegradability, and biocompatibility [12]. Various plant-based proteins, including pea protein [13,14,15,16], mung bean protein [17,18], walnut protein, sunflower protein [9], and rice bran protein [3], have been utilized to enhance Cur solubility in water and improve its bioavailability through complexation.

Coconut (*Cocos nucifera* L.) is an important fruit tree, particularly found in tropical regions. The kernel includes the edible white flesh, which contains 40% moisture, 35.2% fat, and 3.8% protein. It serves as the main source for oil, milk, and cream production. After extracting coconut oil, protein can be obtained from the defatted cake through three methods: chemical, enzymatic, and combined enzyme-chemical processes [19]. The protein content in residues ranges from about 18–25% on a dry basis [20]. Coconut protein shows potential for anti-diabetic and antioxidant activities because of its high arginine level [21,22]. The globulin fraction of coconut protein contains significant amounts of aromatic amino acids (e.g., threonine and phenylalanine) and a high proportion of hydrophobic amino acids. The prolamin fraction also contains aromatic amino acids, which can participate in bonding to Cur and enhance its solubility and antioxidant activity [22]. Recently, numerous studies have been conducted on byproducts of coconut oil, such as the complexation of coconut protein with tannic acid and the noncovalent interactions with coffee polyphenols. These interactions aim to enhance the physicochemical and functional properties of coconut protein. Additionally, researchers have studied the creation of bioactive films made from coconut processing byproducts [23,24,25,26,27,28,29]. To our knowledge, research on coconut protein primarily focuses on these properties, with no studies on its potential as a nanostructure delivery system. Therefore, in this study, coconut protein nanoparticles loaded with Cur were created under acidic conditions (pH 2). Scanning electron microscopy (SEM) was used to observe their morphology, while structures were characterized using fluorescence spectroscopy, FTIR spectroscopy, and XRD analysis to understand how Cur is encapsulated into nanoparticles. Additionally, their radical scavenging activity and release kinetics under simulated gastrointestinal conditions were evaluated.

## 2. Materials and Methods

### 2.1. Materials

Defatted coconut meal from the oil extraction process was purchased from local suppliers in Tehran, Iran. Cur, pancreatin (activity of 275 units/mg), and pepsin (activity > 3000 units/g) were purchased from Bio Basic (Bio Basic Inc., Toronto, ON, Canada). ABTS (2,2′-azino-bis (3-ethylbenzothiazoline-6-sulphonic acid)) was purchased from Sigma-Aldrich (St. Louis, MO, USA). Other chemicals used were of analytical grade.

### 2.2. Extraction of Coconut Protein Concentrate

The extraction of coconut protein concentrate (CPC) followed the procedure described by Rodsamran and Sothornvit [30] with slight modifications. Firstly, the coconut oil cake was finely ground using a water-cooled mill, then mixed with DW at a ratio of 1:12 (*w*/*w*). The pH of dispersion was adjusted to pH 9 with NaOH 2 M and stirred for 2 h, followed by centrifuging at 10,000× *g* (4 °C) for 30 min. The supernatant was obtained and precipitated at pH 4 using 2 M HCl. The protein pellet from the supernatant was then collected. The protein suspension was readjusted to pH 7.0 using 1 M NaOH and dialyzed against deionized water for 24 h and freeze-dried. The Kjeldahl method was used to estimate the protein content. The protein content of the powder was 64%.

### 2.3. Coconut Protein Nanoparticle Formation

CPC was dissolved in DW (containing sodium azide) with a concentration of 50 mg/mL. After stirring this solution for 2 h, it was left overnight in the refrigerator to complete the protein hydration. The pH of the CPC solution was adjusted to 2.0 using 8 M HCl, and then the solution was stirred at 200 rpm at 85 °C for 5 and 48 h. After stopping the nanoparticle process by cooling down using tap water, the pH of the dispersion was adjusted to 7.0 using 5M NaOH. The resultant complex was either stored at 4 °C or lyophilized for further testing. Samples stored at 4 °C were used for Entrapment Efficiency and Loading Capacity, Fluorescence Spectroscopy, Particle size and Polydispersity Index, zeta potential, ABTS Antiradical Activity, and In vitro Controlled Release, while lyophilized samples were used for Scanning Electron Microscopy, FT-IR Spectroscopy, and X-ray Diffraction Spectroscopy.

### 2.4. Scanning Electron Microscopy (SEM)

The morphology observation of the lyophilized nanoparticles was conducted using scanning electron microscopy (FE-SEM Tescan Mira3, Brno, Czech Republic), similar to Yan et al. [31]. Prior to analysis, the samples in powder form were mounted on carbon tape and directly sputter-coated with platinum for 30 s.

### 2.5. Preparation of Cur-Coconut Protein Nanoparticle Complexes

CPC solution with a protein concentration of 50 mg/mL was prepared by dissolving CPC in DW through stirring for 2 h at room temperature. The protein solution was stored at 4 °C for 12 h to hydrate completely. The pH of the protein mixture was then brought to pH 2.0 by adding 8 M HCl. The Cur solution in ethanol (did not exceed over 0.2% (*v*/*v*)) and was added to the protein dispersion or DW with a Cur-protein ratio of 1:50 (mg/mL), followed by stirring at 85 °C for 5 h in a dark place. After stopping the nanoparticle process by cooling down using tap water, the pH of the solution was adjusted to 7.0 with 5 M NaOH. The resultant complex was stored at 4 °C or lyophilized for further testing, as described in Section 2.3.

### 2.6. Measurement of Entrapment Efficiency (EE) and Loading Capacity (LC)

The EE and LC of Cur-loaded CPNP were evaluated by the method of Peng et al. [3]. Briefly, the CPNP-Cur solutions were centrifuged at 10,000× *g* for 30 min. The collected supernatant was diluted with ethanol to reach the desired concentration, and a UV-visible spectrophotometer Shimadzu UV-1800 (Shimadzu Corporation, Kyoto, Japan), was used to determine the absorbance of Cur at 420 nm. The amount of Cur was estimated using a calibration curve of Cur in ethanol solution (R^2^ = 0.999). Equations (1) and (2) were used to determine the EE and LC, respectively:(1)$$\%\mathrm{EE}\;=\;\frac{\mathrm{Total}\;\mathrm{curcumin}-\mathrm{Free}\;\mathrm{curcumin}}{\mathrm{Total}\;\mathrm{curcumin}}\times 100$$(2)$$\%\mathrm{LC}=\frac{\mathrm{Mass}\;\mathrm{of}\;\mathrm{curcumin}\;\mathrm{in}\;\mathrm{nanoparticles}}{\mathrm{Mass}\;\mathrm{of}\;\mathrm{protein}\;\mathrm{in}\;\mathrm{nanoparticles}}\times 100$$

### 2.7. Fluorescence Spectroscopy Analysis

A spectrofluorometer (Cary Eclipse, Palo Alto, CA, USA) was used for evaluating the fluorescence properties. Initially, all samples, including CPC, CPN, and CPNP-Cur, were properly diluted with DW to concentrations of 0.2 mg/mL and 4 µg/mL for protein and Cur, respectively. Cur aqueous dispersion with the same Cur concentration was also prepared at DW for comparison. To determine the intrinsic fluorescence, the samples were excited at 280 nm, and the emission spectra were scanned from 300 to 450 nm. Additionally, the samples were analyzed at an excitation wavelength of 420 nm to investigate the fluorescence properties of Cur, with an emission spectrum recorded from 450 to 700 nm.

### 2.8. Particle Size, Polydispersity Index, and Zeta Potential Measurement

The average particle size distribution, polydispersity index (PDI), and zeta-potential were determined by dynamic light scattering (DLS) using the Zetasizer Nano ZN (Malvern Instruments, Worcestershire, UK) at room temperature, following the method of Han et al. [32]. Before analysis, the sample concentrations were regulated at 5 mg/mL with phosphate-buffered saline (PBS) to reduce the influence of multiple light scattering.

### 2.9. FT-IR Spectroscopy Methodology

FT-IR analysis of the samples was performed following the method of Shakoor et al. [33]. The FT-IR spectra of lyophilized samples were recorded using a Bruker FT-IR Spectrum 1 (Billerica, MS, USA). The samples were prepared using the KBr disc method and scanned from 4000 to 500 cm^−1^ at specific resolutions.

### 2.10. X-ray Diffraction Spectroscopy (XRD)

The X-ray patterns of the coconut protein nanoparticles were obtained with slight modifications to the method proposed by Ren et al. [13]. The molecular arrangements of Cur, CPC, CPN, and CPNP-Cur were obtained using a Philips PW1730 X142 X-ray diffractometer (PANalytical, Almelo, The Netherlands) using Cu Kα radiation. Samples were scanned constantly over a 2θ angle range from 5 to 50° with a step size of 0.05°.

### 2.11. ABTS Radical Scavenging Assay Method

ABTS radical scavenging was undertaken following the method of Blanco-Padilla et al. [34]. Initially, the ABTS^+^ was prepared by the interaction of 7.4 mM ABTS in PBS (200 mM, pH 7.4) and 2.6 mM potassium persulfate, followed by incubation for about 18 h at 25 °C. The ABTS^+^ solution was then adjusted to an absorbance of 0.7 ± 0.1 at 734 nm by diluting with DW. In all samples, Cur and protein were adjusted to the same concentrations of 0.1 mg/mL and 5 mg/mL, respectively. Then, 50 µL of the samples was added to 1.0 mL of ABTS radical solution and then stored in a dark place for 10 min. The absorbance (A) of samples was recorded at 734 nm, and DW was used as the control. The antiradical activity was estimated as Equation (3):(3)$$\%\;\mathrm{ABTS}\;\mathrm{radical}\;\mathrm{scavenging}\;\mathrm{activity}\;=\;\frac{A\_\mathrm{control}-A\_\mathrm{sample}}{A\_\mathrm{control}}\times 100$$

### 2.12. In Vitro Controlled Release

The in vitro controlled release of Cur from CPNP-Cur was performed in a simulated gastric fluid with a pH of 1.2, followed by a simulated intestinal fluid with a pH of 7.5, based on the method of Mohammadian et al. [18]. Equal volumes of CPNP-Cur and SGF were loaded into a dialysis bag with a 12,000 Da molecular weight cut-off. The bag was then immersed in a Pyrex lab bottle containing a 1:1 mixture of ethanol and enzyme-free SGF (150 mL total) and incubated for 2 h at 37 °C with constant agitation at 100 rpm. To simulate the intestinal environment, the pH of the dialysis contents was raised to 7.5 to inactivate pepsin, and 6 mL of SIF was added. Finally, the bag was transferred to a medium consisting of a 1:1 mixture of ethanol and enzyme-free SIF and incubated for 4 h at 37 °C with continuous agitation at 100 rpm. At regular time intervals, 2 mL aliquots were taken from each released medium and replaced with the same volume of fresh medium. The amount of Cur was estimated using a Cur calibration curve in the same release environment. Cumulative Cur release from CPNPs was assessed by plotting the calculated cumulative release percentages (%) according to Equation (4).(4)$$\%\;\mathrm{Cumulative}\;\mathrm{Cur}\;\mathrm{Release}=\frac{\mathrm{Amount}\;\mathrm{of}\;\mathrm{released}\;\mathrm{curcumin}\;\mathrm{in}\;\mathrm{solution}}{\mathrm{initial}\;\mathrm{curcumin}\;\mathrm{concentration}}\times 100$$

### 2.13. Mathematical Modeling and Drug Release Kinetics

To evaluate the Cur release kinetics from CPNPs, four mathematical models were used, including zero order, first order, Higuchi Model, and Korsmeyer-Peppas model. The R^2^ value was calculated to indicate the best-fit model. The value of ‘n’ represents the slope obtained in the Korsmeyer Peppas graph and predicts the Cur release mechanism as follows [35]:
n ≤ 0.45 Fickian diffusion0.45 < n< 0.89 Anomalous transportn > 0.89 Super case II transport mechanism (erosion)

### 2.14. Statistical Analysis

All data were obtained in triplicate, and statistical analyses were performed using one-way ANOVA performed by SPSS software version 16 (IBM software, Armonk, NY, USA). Duncan’s multiple comparison test (*p* < 0.05) was conducted to evaluate significant differences between samples.

## 3. Results and Discussion

### 3.1. The Fabrication of Coconut Protein Nanoparticles and Characterization Using Scanning Electron Microscopy

To decrease environmental pollution and improve the value of coconut meal, coconut protein was extracted using an alkaline extraction method and then used to prepare nanoparticles through a self-assembly process. Although this study aimed to synthesize nanofibrils from coconut protein, after adjusting the pH to 2, followed by heating at 85 °C for 5 h according to Mohammadian et al. [18], the images obtained from SEM revealed nanoparticles in smooth, round shapes. The samples were taken at different time intervals. As shown in Figure 1a,b, the surface morphology of the fabricated nanoparticles during 5 h and 48 h was mostly uniform, with round globular shapes that had an approximate size of 80 nm. Based on these results, the nanoparticles fabricated in 5 h were selected for Cur encapsulation and further analysis. There were other methods of fabrication of nanoparticles from plant protein, such as the fabricated soybean protein nanoparticles developed by Yuan et al. [36] through self-assembly of the amphiphilic hydrolysate after partial hydrolysis of protein using pepsin and pancreatin, which achieved nanoparticles with a size of 82 nm. Perovic et al. [37] prepared nanoparticles from pumpkin leaf protein via heat treatment at 90 °C at a pH of 9.3 for 20 min. They obtained nanoparticles with a size of approximately 18 nm and 21 nm from protein extracted by the conventional alkaline method and enzyme-assisted extraction, respectively. Also, the curcumin exhibited limited bioavailability (approximately 1%) due to its poor water solubility, crystalline structure (with mean particle sizes of 0.4 µm and 30.8 µm), and chemical degradability [38]. Therefore, in this study, the complexation of curcumin with coconut protein nanoparticles reduced its crystallinity, which improved its bioavailability. Then, coconut protein nanoparticles were employed as a carrier of curcumin to enhance its solubility and bioavailability.

### 3.2. Entrapment Efficiency (EE) and Loading Capacity (LC)

The EE and LC of Cur in the CPNPs were obtained by calculating the Cur concentration that existed in the supernatant after the encapsulation process [3]. The EE and LC of Cur in CPNP were 96.6% and 19.32 µg/mg protein, respectively. These results indicated the high potential of nanoparticles in interaction with Cur due to the highly hydrophobic surface [39]. Numerous plant-based nanoparticles have been developed to encapsulate Cur, aiming to increase its stability, bioavailability, and therapeutic efficacy. A comparison between the EE and LC of Cur in the CPNPs and those of other plant protein nanoparticles is represented in Table 1. The EE of CPNPs was significantly (*p* < 0.05) higher than the EE of walnut protein, up to 61.45% [40]. Our results indicated that the EE of Cur is superior to that of other plant-based nanoparticles, such as nanoparticles fabricated from rice bran waste protein, which can encapsulate Cur up to 93.56% [3], and sunflower protein, which can encapsulate Cur up to 83% [41]. In comparison to our results, Xu et al. [42] reported a significantly lower percentage of EE (about 65%) and a higher LC (12.6%) of Cur in the rice protein nanoparticles. They reported that the rice protein nanoparticles increased LC but decreased EE of Cur as the Cur concentration increased. Similarly, Du et al. [43] indicated that the curcumin loading capacity of soy protein nanoparticles increased from 6.38 µg/mg protein to 26.76 µg/mg protein by increasing curcumin concentration. The LC of curcumin in CPNPs was significantly higher than that of encapsulated pea protein nanoparticles reported by Shakoor et al. [33], which was about 0.32% due to the high polymer-to-curcumin ratios applied in the encapsulated particles. Zhang et al. [44] revealed that the LC could be enhanced from 0.62% to 4.38% by reducing the mass ratio. Guo et al. [45] also reported that using more hydrophilic protein in the complexation of curcumin resulted in high EE and a low percentage of LC.

### 3.3. Fluorescence Spectroscopy

Fluorescence spectroscopy is the main approach for the analysis of interactions between proteins and ligands [49]. Fluorescence spectra of CPC, CPNP, and CPNP-Cur at the Cur excitation wavelength are illustrated in Figure 2a. The aqueous solution of Cur indicated a minimum intensity broad peak at 493 nm. In contrast, fluorescence intensity increased significantly after binding to CPNP. These results demonstrate the transformation of Cur from a hydrophilic medium to a less hydrophilic environment, which is related to the hydrophobic interactions between Cur and hydrophobic residues of CPNP. Fluorescence spectroscopic studies of protein-Cur interactions, including pea protein [50], walnut protein [40], rice protein [42], and zein [51], have indicated that the aromatic amino acids induce hydrophobic interactions between protein and Cur. The intrinsic fluorescence of samples was also studied to evaluate the alteration in the tertiary structure of CPC and the CPNP-Cur. The resulting spectra are demonstrated in Figure 2b. The fluorescence intensity significantly decreased after the fabrication of nanoparticles, which represents the change in the spatial structure of the CPC. A higher level of quenching was observed in CPNP-Cur, which is related to the potential of Cur in fluorescence quenching. Dong et al. [52] also reported that there was no blue or red shift in the encapsulation of Cur into egg white protein isolate. However, the fluorescence intensity decreased significantly after Cur encapsulation. Weng et al. [49] fabricated Pseudostellariae protein nanoparticles via heat treatment at 100 °C for 30 min, followed by pH adjustment to 5.70. They indicated that the intrinsic fluorescence of the protein nanoparticles decreased after the encapsulation of Cur concentration. They also showed that through the increase in Cur concentration, the fluorescence of accessible tryptophan residues was quenched, resulting in a significant reduction in fluorescence intensity.

### 3.4. Average Particle Size and Polydispersity Index (PDI) and Zeta Potential

The average particle size and polydispersity of samples were demonstrated in Figure 3. The particle size of CPC was about 67.77 ± 1.10 nm with a PDI of 0.58. These amounts increased for CPNPs to 111.76 ± 0.85 nm and 0.4, respectively. Although the CPNPs indicated a larger size, it was more homogenous than CPC particles. Yuan et al. [36] also reported that, during nanoparticle construction of soy protein, the average particle size increased from about 68 nm to 82 nm, but created more homogenous particles with lower amounts of PDI. Perović et al. [37] also reported that the size of nanoparticles produced through heat treatment was larger than native pumpkin leaf protein. They attributed this observation to heat treatment, which induced partial unfolding of the protein because of the denaturation. So, previously buried hydrophobic residues became exposed, promoting hydrophobic interactions and leading to protein aggregation. After adding Cur, the average particle size was approximately 94.04 ± 1.97 nm, which could be related to the incorporation of Cur into the internal hydrophobic core of the structure [43]. This result aligns with the findings of Guo et al. [45], who demonstrated that the addition of Cur to pea protein isolate resulted in smaller particle sizes compared to pea protein alone. This reduction in size can be attributed to the hydrophobic nature of curcumin, which led to the expulsion of water from the complex. Consequently, this process resulted in a more compact structure with smaller particles. However, PDI increased to 0.9 in CPNP-Cur. This result was consistent with those reported by Ghobadi et al. [53], who encapsulated Cur in nanoparticles fabricated using Grass pea protein isolate and Alyssum homolocarpum seed gum. They indicated that the hydrophobic interaction between the Grass pea protein and Cur resulted in an increase in particle size and PDI of about 535 nm and 1.85, respectively. The zeta potential of the samples was shown in Figure 3b. The negative charge was observed in all samples because the pH of the analyzed solution was remarkably higher than the isoelectric point of CP (pI 4) [54]. The zeta potential of CPC was −22.4 ± 0.20 mV, which significantly enhanced (*p* < 0.05) to −24.93 ± 0.13 mV after formation of CPNPs. This phenomenon can be related to the unfolding of proteins at high temperatures, which leads to a greater presence of negative charge on the surface of the proteins. This result agreed with Qiao et al. [14], who reported a higher magnitude of zeta potential through further increasing the temperature at high pH. After the addition of Cur, the zeta potential increased to −26.92 ± 0.12 mV, because curcumin has a certain negative charge (about 11.7 mV), which provided a larger zeta potential in CPNP-Cur [51]. Previous studies have suggested that a high zeta potential of more than 30 mV is essential for achieving strong colloidal stability [49]. Therefore, we hypothesize that hydrogen bonding and hydrophobic interactions play a crucial role in stabilizing Cur-loaded nanoparticles. This result corresponds to the data obtained from FTIR and fluorescence analysis. This result was consistent with the findings of Yan et al. [46], who reported that the complexation of soy protein isolate with Cur led to the formation of larger particles, despite a high magnitude of zeta potential. They also attributed this phenomenon to the stronger crosslinking caused by hydrogen bonding and hydrophobic interaction between the protein and Cur, which surpasses the effects of electrostatic repulsion.

### 3.5. FT-IR Spectroscopy

The potential intermolecular interactions of CPC, Cur, CPNP, and CPNP-Cur were characterized by FT-IR spectra (Figure 4). The spectrum of CPC showed prominent absorption peaks at 3295.13, 2925.12, 1646.31, and 1541.18 cm^−1^, which were associated with amide A, amide B, amide I, and amide II, respectively [14]. In the spectrum of Cur, the band at 3456 cm^−1^ corresponded to the hydroxyl group, which is related to the tensile vibration of the phenolic hydroxyl group. Several characteristic bands were observed mainly in wave numbers 1628.01, 1510.33, 1428.80,1280.29, and 1153.96 cm^−1^, related to the stretching vibration of inter-ring chains of ketone and aromatic rings, consistent with the results of previous studies [55]. Most of these bands disappeared after the complexation by CPNP. The limited bending and stretching of Cur molecules after binding to CPNP revealed the great complexation of Cur to nanoparticles [45]. Mohammadian et al. [56] reported a similar phenomenon, where the characteristic bands of Cur disappeared upon binding to whey protein, indicating that hydrophobic interaction results from the interaction between the tryptophan residues in CPNP and the aromatic rings in Cur. The red shift was observed in Amide I in CPNP and CPNP-Cur related to the change in conformations of proteins’ secondary structures. Compared to CPC, the characteristic peak of the amide II band in CPNP and CPNP-Cur showed a red shift and a blue shift, respectively. These results demonstrated that heating and pH-shifting treatments could alter the N–H or C–N bond of CPC, consistent with previous reports [14,57]. The nanoparticle fabrication and complexation with Cur resulted in a peak displacement from 2925.12 cm^−1^ to 2924.74 cm^−1^ for CPNP and from 2925.12 cm^−1^ to 2924.98 cm^−1^ for CPNP-Cur, corresponding to the CH, CH2, and CH3 interaction bonds. There was also a significant red shift in amide A. For CPC, peaks were observed at 3295.13 cm^−1^, whereas they appeared at 3390 cm^−1^ and 3391 cm^−1^ in CPNP and CPNP-Cur, respectively. These results suggested that hydrogen bonds also participate in CPNP and Cur complexation due to the presence of hydroxyl and carbonyl groups in Cur. Chen et al. [27] reported that the coconut globulin-tannic acid complexation shifted from 3353 cm^−1^ to 3423 cm^−1^, which is attributed to hydrogen bonding between the particles.

### 3.6. X-ray Diffraction Analysis

The physical phase of the samples was analyzed using X-ray diffraction. As depicted in Figure 5, the XRD patterns of Cur in pure form indicated multiple distinctive peaks at 2θ of 8.9°, 12.2°, 14.4°, 17.2°, 18°, 23.3°, 24.4°, and 25.4°, confirming its highly crystalline nature [48]. CPC indicated a broad peak at about 20° related to the amorphous nature of native proteins [58]. In the XRD pattern of Cur-CPNP, peaks became weak, suggesting that the Cur existed in an amorphous state; new peaks at about 64° and 72° emerged, which referred to the existence of salt crystals resulting from pH adjustment [1,59]. Yuan et al. [60] also showed that the XRD spectrum of physically bonded Cur to soy protein preserved the crystalline structure. However, these peaks faded in the spectra of soy protein nanoparticles, which indicated a successful encapsulation of Cur and a transition from a crystalline form to an amorphous state. Since amorphous forms generally exhibit higher bioavailability than their crystalline counterparts. Therefore, the transformation of Cur’s crystalline structure upon complexation with nanoparticles might enhance its uptake [1].

### 3.7. ABTS Antiradical Activity

Cur contains a high antioxidant capacity. However, its low solubility in aqueous solution reduces this potential. Previous studies suggest that to improve Cur’s antioxidant activity, it must be encapsulated in lipophilic polymer matrices [61]. The ABTS scavenging method was used to evaluate the antiradical activity of CPC, CPNP, CPNP-Cur, and free Cur. Figure 6 shows the antiradical activity of samples with the same Cur concentration (1 mg/mL). The antiradical activity of Cur in an aqueous medium was significantly limited, with only 10% effectiveness. The antiradical activity of the CPNP-Cur was about 5.5 times higher than that of free Cur. CPC antiradical activity decreased after conversion to nanoparticles, which could be related to changes in the protein structure, such as denaturation. These results demonstrated that the CPNPs are suitable carriers for lipophilic compounds such as Cur in aqueous solution. Zhang et al. [62] reported that the antioxidant activity of Cur was significantly increased after the encapsulation into sodium caseinate and zein nanoparticles, which was attributed to the increased solubility of Cur in water. Niu et al. [63] confirmed that Cur-loaded ovalbumen-carboxymethyl cellulose nanoparticles significantly improved their functional properties via increasing antioxidant activity.

### 3.8. In Vitro Release Studies

Cur release from CPNPs under a simulated gastrointestinal environment is shown in Figure 7. The release of Cur from CPNPs at the end of the incubation in a gastric fluid was 9.505%, which indicates very low or no release at all during the 2 h of residence time. Cur encapsulated in CPNPs indicated a lower release rate, which demonstrated an effective improvement in Cur stability [46]. This is likely due to the entrapment of Cur in the hydrophobic protein cavity, which protects it from degradation [64]. Meng et al. [65] achieved a cumulative Cur release of about 6% from whey protein coated with Octenyl succinic anhydride debranched starch particles after 1 h in SGF. They attributed this result to the ability of particles to protect Cur from pepsin. This is due to Cur being located in the hydrophobic part of the particles, which decreases pepsin’s access to Cur because of the steric hindrance and van der Waals forces. After transferring nanoparticles from SGF to SIF, no abrupt release was observed, and it continued at a constant rate, like that in the SGF. Huang et al. [66] encapsulated resveratrol in nanoparticles composed of zein and pectin and evaluated their antioxidant capacity in the simulated gastrointestinal tract. They demonstrated that particles with a high specific surface area promoted resveratrol release in the gastrointestinal environment. Therefore, the larger size and dense structure of CPNP-Cur could restrict the pancreatin activity and result in the sustained release of Cur. The steady release of Cur continued in the SIF medium, and the percentage of the cumulative release increased up to 29.2% after 4 h. This also indicates that CPNPs could be used as a targeted delivery carrier for hydrophobic compounds. In the study of Gupta and Ghoshal [67], the Cur release from green gram (mung bean) protein hydrogel showed similar trends. They reported that the increase in Cur release in the intestinal conditions refers to the enhanced solubility in the alkaline medium. Under alkaline conditions, Cur becomes hydrophilic due to the deprotonation of hydroxyl groups. On the other hand, the nanoparticles gradually disintegrated over time when exposed to potassium dihydrogen orthophosphate, and the water permeating into the structure allowed Cur to diffuse into the SIF medium. Meiguniet et al. [17] calculated the Cur release from Cur-loaded Succinylated mung bean protein and Arabic to be approximately 48% and 70% at SGF and SIF, respectively, which were significantly higher than our findings. They reported that pepsin and pancreatin enzymes play a critical role in the destruction of the electrostatic bond in the complex coacervate structure.

### 3.9. Mathematical Modeling and Drug Release Kinetics

The in vitro release data of Cur from CPNPs was analyzed by fitting it into four mathematical models as depicted in Figure 8. The release mechanism describes how an active substance is released from the carrier matrix, typically involving multiple processes (including diffusion, dissolution, swelling, relaxation of polymeric chains, erosion, and degradation) [68]. Table 2 indicates the mechanism of Cur release based on the ‘n’ value achieved from the Korsmeyer–Peppas model and R^2^ values using linear regression. Zero-order is the perfect model followed for the release of drugs from nanoparticles [69]. Comparing the R^2^ values, the Cur release data fitted well to zero-order, first-order, and Korsmeyer-Peppas models with high correlation coefficients. The k and n values for the Korsmeyer–Peppas model were 0.05 and 1.05, respectively. The n values > 0.89 referred to super case II, which involved disentanglement and erosion of the nanoparticles [70]. Ganguly et al. [68] evaluated the potential of BSA nanoparticles for encapsulating the hydrophobic compound garcinol A and improving its bioavailability. They reported that the n value in all samples was >1, which related to the role of the macromolecular polymeric chain relaxation.

## 4. Conclusions

This research indicated the ability of CPC to convert into nanoparticles with a size of about 80 nm through heating at a pH of 2.0. The CPNP was used as a delivery system of curcumin (Cur) and obtained high EE and LC values compared with other plant protein nanoparticles. Fluorescence spectroscopy and FTIR data showed that hydrophobic interactions and hydrogen bonding played a critical role in the complexation of Cur-coconut protein nanoparticles. The complexation of Cur nanoparticles was shown to have significant radical scavenging activities compared to Cur solubilized in water. Cur release was steady and sustained even in the gastric medium. Overall, our findings showed that coconut protein nanoparticles were suitable nanocarriers for Cur due to improved antioxidant activity and controlled release behavior. These results are valuable for the development of coconut protein nanoparticles to use as a novel nano-delivery system of bioactive components.

## Figures and Tables

**Figure 1 pharmaceutics-17-01247-f001:**
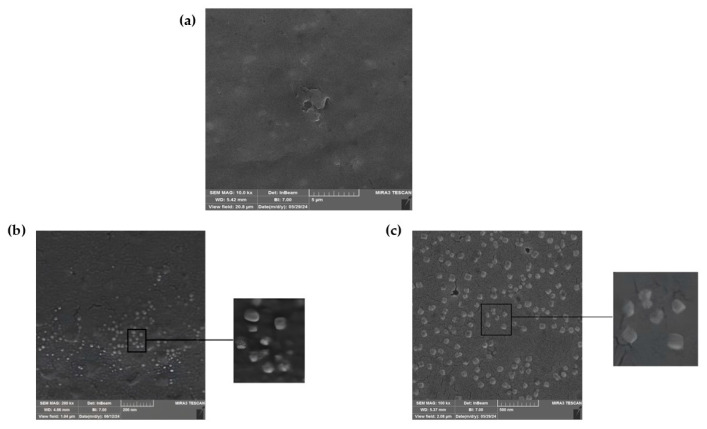
(**a**) SEM images of coconut protein concentrate (CPC); (**b**) coconut protein nanoparticle heated for 5 h (CPNP5); (**c**) coconut protein nanoparticle heated for 48 h (CPNP48).

**Figure 2 pharmaceutics-17-01247-f002:**
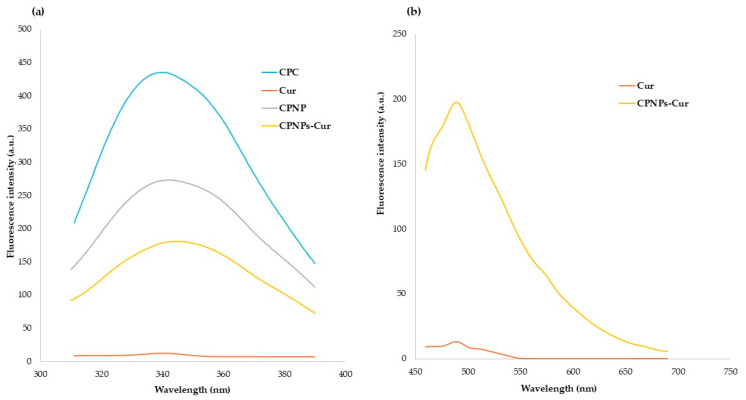
(**a**) Fluorescence emission spectra of various samples at 480 nm. (**b**) at 280 nm, an Aqueous solution of Cur with a concentration of 4 µg/mL (Cur), coconut protein concentrate (CPC), coconut protein nanoparticle (CPNP), and the complexation of coconut protein nanoparticle with Cur with a curcumin concentration of 4 µg/mL (CPNPs-Cur).

**Figure 3 pharmaceutics-17-01247-f003:**
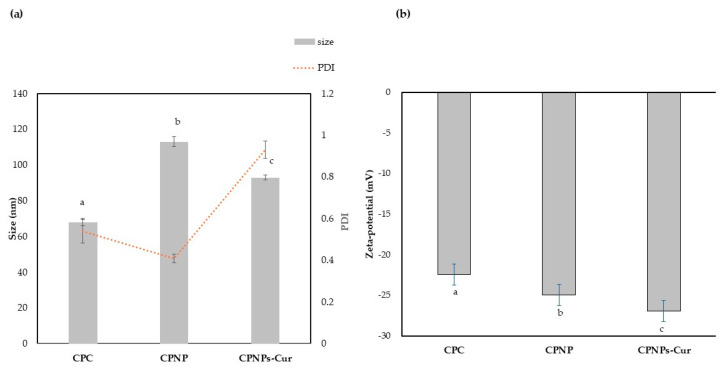
(**a**) Droplet size and PDI (**b**) Zeta-potential. coconut protein concentrate (CPC), coconut protein nanoparticle (CPNP), and the complexation of coconut protein nanoparticle with Cur (CPNPs-Cur). Data are expressed as mean ± standard deviation (*n* = 3). Different superscript letters (a, b, and c) indicate significant differences (*p* < 0.05) according to Duncan’s multiple comparison test.

**Figure 4 pharmaceutics-17-01247-f004:**
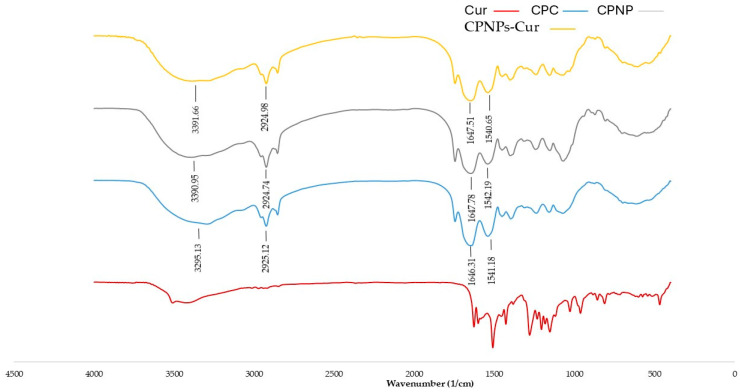
FTIR spectra of aqueous solution of Cur (Cur), coconut protein concentrate (CPC), coconut protein nanoparticle (CPNP), and the complexation of coconut protein nanoparticle with Cur (CPNPs-Cur).

**Figure 5 pharmaceutics-17-01247-f005:**
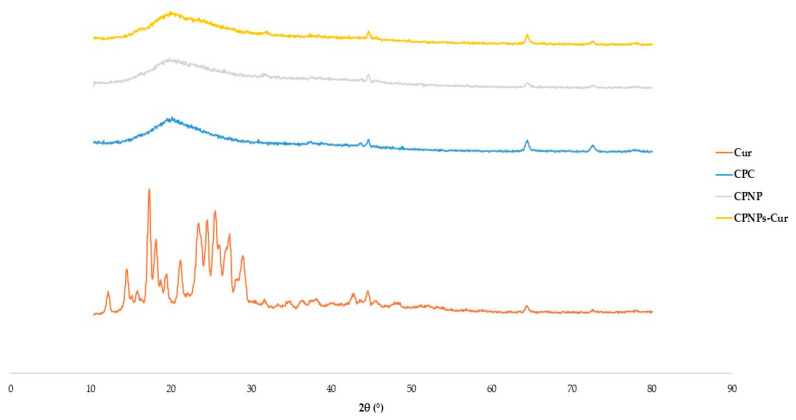
XRD patterns of aqueous solution of Curcumin (Cur), coconut protein concentrate (CPC), coconut protein nanoparticle (CPNP), and the complexation of coconut protein nanoparticle with Cur (CPNPs-Cur).

**Figure 6 pharmaceutics-17-01247-f006:**
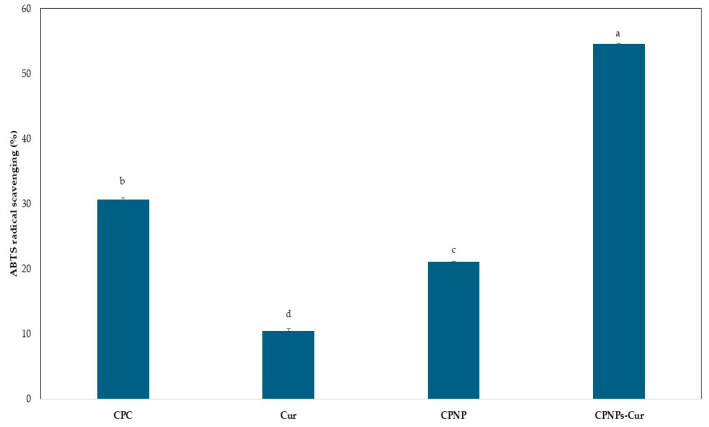
ABTS- the radical scavenging activity of different samples, including the aqueous solution of Curcumin with the concentration of 0.1 mg/mL (Cur), coconut protein concentrate (CPC), coconut protein nanoparticle (CPNP), and the complexation of coconut protein nanoparticle with Cur (CPNPs-Cur). Data are expressed as mean ± standard deviation (*n* = 3). Different superscript letters (a, b, c, and d) indicate significant differences (*p* < 0.05) according to Duncan’s multiple comparison test.

**Figure 7 pharmaceutics-17-01247-f007:**
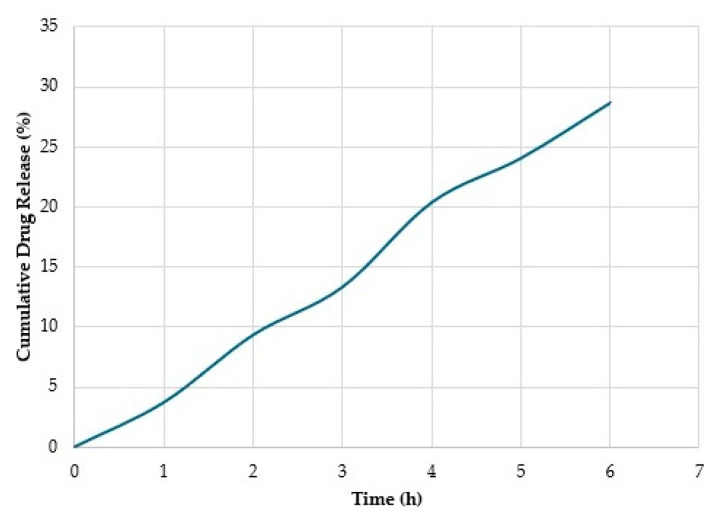
In vitro release of Cur from CPNPs. Blue line represents drug release profile from the nanoparticles, indicating an increase in drug release progressively from 0% at time 0 up to about 29% at 6 h, demonstrating a sustained release characteristic from the CPNPs.

**Figure 8 pharmaceutics-17-01247-f008:**
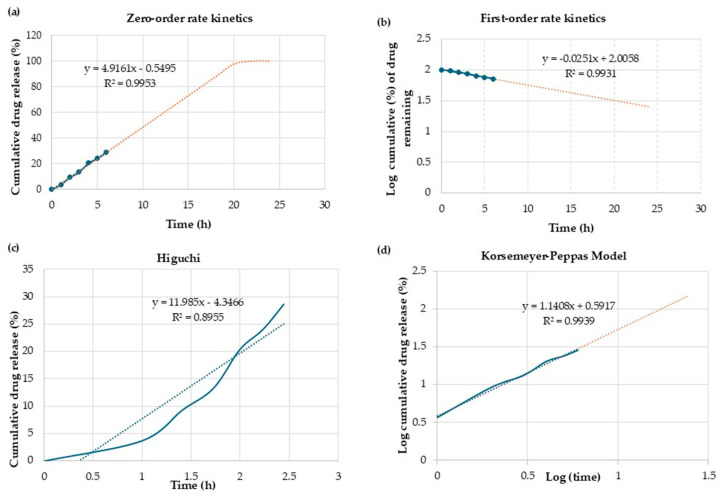
Cur release from CPNPs fitted with (**a**) zero-order rate kinetics, (**b**) first-order rate kinetics, (**c**) Higuchi model, (**d**) Korsmeyer-Peppas model. The blue dots represent the experimental data points of cumulative drug release (%) measured at each time point (hours). The solid blue line connects the experimental data points, visually showing the progression of drug release over time. The blue and yellow dotted line represents the linear regression fit of the data according to zero-order kinetics.

**Table 1 pharmaceutics-17-01247-t001:** Comparison of the various characteristics of plant protein nanoparticles used for curcumin complexation.

Plant Protein Source	Curcumin-Loaded Nanoparticle Size (nm)	EE (%)	LC (%)	References
Coconut protein nanoparticles	94	96.6	1.932	(This study)
Soybean protein nanoparticle (coating by dextran dialdehyde)	199.2	92.37	0.924	[46]
Zein (complexed with Chondroitin sulfate)		94.7	3.8	[47]
Pea protein nanoparticle	867.4	93	0.32	[33]
Pea protein nanoparticle (using different surfactants)	160.23–191.20	89.91–93.69	~0.5–1.5	[45]
Rice protein hydrolysate	132.16–150.28	55.5	5.6	[48]
Rice bran waste	384	93.56	26.20	[3]
Polyelectrolytic complex of acylated rapeseed cruciferin and chitosan	217–454	72–79	5.4–6.2	[33]
Sunflower protein nanoparticle	174.64	95.4		[9]

**Table 2 pharmaceutics-17-01247-t002:** Correlation coefficient values for determination of drug release kinetics.

Sample	Zero-Order	First-Order	Higuchi	Korsmeyer Peppas	
	R^2^	R^2^	R^2^	R^2^	*n*
CPNP-Cur	0.99	0.99	0.89	0.88	1.05

## Data Availability

The original contributions presented in this study are included in the article. Further inquiries can be directed to the corresponding author(s).

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
