# Peer review of "Fabrication and Characterization of Curcumin-Complexed Nanoparticles Using Coconut Protein Nanoparticles"

_pharmaceutics, 2025, doi:10.3390/pharmaceutics17101247_

Round 1
Reviewer 1 Report
Comments and Suggestions for Authors
The manuscript entitled “Fabrication and characterization of curcumin-complexed nanoparticles using coconut protein nanoparticles” presents a simple idea for obtaining nanoparticles. In order to be accepted for publication, this manuscript needs to be improved. Below are the points that need to be improved.
The introductory part is too brief and without clearly mentioning what is new in this study. What is the applicability of this system? Please improve the introductory part of the manuscript.
The authors said on line 46: “natural polymers, such as lipids”. Lipids are not natural polymers. Please modify.
2.5. Preparation of Cur-Coconut Protein Nanoparticle Complexes
The authors said:“CPC solution with a protein concentration of 50 mg/mL was prepared by dissolving CPC in DW through stirring for 2 h at room temperature.” What is the total amount of Coconut protein concentrate (CPC) used in this study? Please add a table with the experimental program in the manuscript specifying the exact amounts of the components used in the experiments. Were the studies done on a single sample? Wasn't an experimental program followed in which several parameters were varied?
The authors mention: “The Cur solution in ethanol (did not exceed over 0.2% (v/v)) 119 and was added to the protein dispersion or DW with Cur-protein ratio of 1:50 (mg/mL), 120 followed by stirring at 85 °C for 5 h in a dark place.” What is the amount of curcumin used in this study?
2.12. In vitro Controlled Release
The authors mention that they performed the release test using a mixture of ethanol and SGF/SIF medium. Please explain why you used ethanol and what is the relevance for using this medium in the release tests. Normally, solutions that mimic the physiological environment should be used for the release test.
3.2. Entrapment Efficiency (EE) and Loading Capacity (LC)
The authors said that: “The EE and LC of Cur in CPNP were 96.6 % and 19.32 µg/mg protein, respectively. It is necessary to mention the initial amount of Cur introduced into the system. Also, they mentioned that “the high potential of nanoparticles in interaction with Cur due to the highly hydrophobic surface”. Please improve the discussion and present bibliographical references that support your statements.
3.4. Particle Size and Polydispersity Index (PDI)
The authors mentioned that the particle size distribution was determined by dynamic light scattering (DLS). Please insert in the manuscript the graph resulting from the measurement performed on the Malvern Instruments.
It is observed that the polydispersity index (PDI) of the analyzed samples is quite high, which means either that there is an increased tendency for agglomeration or that there are several populations of particles with different sizes. Please explain.
3.8. In Vitro Release Studies
The authors say that they performed this test in a simulated gastrointestinal environment. Once ethanol was introduced into the release medium, a forced release of the active principle was achieved. To perform a correct release test, an alternative would be for the dialysis bag to be immersed in a solution of PBS and enzyme-free SGF. Please redo the comment or redo the test in an environment that correctly simulates the gastrointestinal environment.
Another observation is that after 6 hours of exposure it was found that the degree of release of curcumin is below 30%. Why did the authors choose to stop the experiment after 6 hours? Please explain!
Reviewer 2 Report
Comments and Suggestions for Authors
The manuscript of the article prepared by Leila Ziaeifar, Maryam Salami, Gholamreza Askari, Zahra Emam-Djomeh, Raimar Loebenberg, Michael J. Serpe and Neal M. Davies “Fabrication and characterization of curcumin-complexed nanoparticles using coconut protein nanoparticles” described studies regarding preparation and characterisation of coconut protein nanoparticle-Curcumin composition.
The manuscript is well-written and would be interesting for the experts of related fields, however, some issues needed to be added and addressed before publication.
- Please specify the use of the term antioxidant activity when referring to the ABTS test. The term antioxidant is broader, while antiradical represents a specific case within it. All antiradicals are antioxidants, but not all antioxidants act through an antiradical mechanism. In this case, you can be certain that the composition demonstrated antiradical activity.
- In subparagraphs 2.3. and 2.5. authors wrote “The resultant complex was kept at 4°C or lyophilized for further testing”. Please specify which sample was used for following experiments.
- It is recommended that paragraph 3 be titled “Results and Discussion.”
- Please improve the quality of Fig.1. Could you enlarge and highlight a fragment of the images to improve visibility of particles?
- Please clarify the importance of reporting decimal places in particle size estimation. Why are some particle sizes reported as whole numbers while others include two decimal places? What is the corresponding error interval or uncertainty associated with these measurements in each case?
- Please specify what type of nanoparticle diameters were measured. If these are average diameters, it would be important, given your PDI values, to also report the diameters of the different fractions and their percentage composition in the sample.
- It is not specified anywhere what the lowercase letters listed in the ABTS activity column mean (Fig. 6). What standard do you use to validate the ABTS test method?
- The paragraph conclusions should be marked as No 4 instead 5.
Consequently, I do recommend accepting this manuscript for publication with major revision.
Reviewer 3 Report
Comments and Suggestions for Authors
The article describes the synthesis and application of curcumin based nanoparticles. The topic of article is new and promising. Many researchers are studying nanoparticles to apply them for biomedical research. The article is well written and esy to read and understand. The results are correctly done and well described using 8 figures and two tables. The article was very well discussed using 71 references. In general, the main aim of article - to present new kind of nanoparticles as antioxidants - was reached. As for English, I am not native speaker, for me English is acceptable. Everything is clear to read and understand.
I have minor remarks only:
1) Table 1. Please, idicate the erros in your onw data because they are known ( Fig. 3).
2) Figure 2. Please add concentrations of pure Cur and Cur in composition of NP. Are they corresponding?
3) Please, add the data on zeta-potential of nanoparticles (you can obtain it from zeta-studies).
4) Fig 6. - the same as for Fig. 2 - Which concetrations of cur and Cur-NP were used?
5) Figure 7. In vitro release of Cur from CPNPs. While 6 hours only were used? Please, prolong to 24 hours.
Round 2
Reviewer 1 Report
Comments and Suggestions for Authors
The authors have improved the manuscript by responding positively to the suggestions I made. Consequently, the manuscript entitled "Fabrication and characterization of curcumin-complexed nanoparticles using coconut protein nanoparticles" can be accepted for publication in this form.
Author Response
Thank you
Reviewer 2 Report
Comments and Suggestions for Authors
Thank you for your corrections.
However, it is desirable the paragraph conclusions should be marked as No 4 instead of 5.
Author Response
We have marked conclusions 4 as suggested for enhanced clarity